



# Intraseasonal variability of the South Vietnam Upwelling, South
# China Sea: influence of atmospheric forcing and ocean intrinsic
# variability.
Marine Herrmann[1*], Thai To Duy[2], Claude Estournel[1]
[1] Université de Toulouse, LEGOS (IRD/CNES/CNRS/UPS), Toulouse, France, 31400 Toulouse, France
[2] Institute of Oceanography (IO), Vietnam Academy of Science and Technology (VAST), Nha Trang, Vietnam
*Correspondence to :* Marine Herrmann (marine.herrmann@ird.fr)
**Short summary**
The South Vietnam Upwelling develops in summer along and off the Vietnamese coast. It brings cold and nutrient-
rich waters to the surface, allowing photosynthesis essential to marine ecosystems and fishing resources. We show
here that its daily variations are mainly due to the wind, thus predictable, in the southern shelf and coast regions.
However, they are more chaotic in the offshore area, and especially in the northern area, due to the influence of
eddies of a highly chaotic nature.





**Abstract**
The South Vietnam Upwelling (SVU) develops off the Vietnamese coast (South China Sea) during the southwest
summer monsoon over four main areas: the northern coastal upwelling (NCU), the southern coastal upwelling
(SCU), the offshore upwelling (OFU) and the shelf off the Mekong River mouth (MKU). An ensemble of ten
simulations with perturbed initial conditions were run with the fine-resolution SYMPHONIE model (1 km
inshore) to investigate the daily to intraseasonal variability of the SVU and the influence of the ocean intrinsic
variability (OIV) during the strong SVU of summer 2018.
The intraseasonal variability is similar for SCU, MKU and OFU, driven to the first order by the wind variability.
MKU and SCU are induced by stable ocean dynamics (the northeastward then eastward boundary current) and
have very little chaotic variability. The OIV has a stronger influence on OFU. In July, OFU mainly develops along
the northern flank of the eastward jet. The influence of OIV is strongest and related to the chaotic variability of
the meridional position of the jet. In August this position is stable and OFU develops mainly in the area of positive
wind curl and cyclonic eddies north of the jet. The influence of OIV, weaker than in July, is related to the
organization of this mesoscale circulation. NCU shows a completely different behavior from the other areas. In
the heart of summer, the large-scale circulation formed by the eastward jet and eddy dipole is well established
with an alongshore current preventing NCU. In early and late summer, the large-scale circulation is weaker,
allowing a mesoscale circulation of strongly chaotic nature to develop in the NCU area. During those periods, the
OIV influence on NCU is very strong and related to the organization of this mesoscale circulation: NCU is favored
(annihilated) by offshore-oriented (alongshore) structures.

**1. Introduction**
The summer general circulation in the central South China sea is largely induced by the prevailing southwest
monsoon winds over the region (Wang et al., 2004; Wyrtki et al., 1961). It is characterized by the development
off the central Vietnamese coast of an anticyclonic gyre (AC) in the south and a cyclonic gyre (C) in the north
(forming an eddy dipole referred to as the ACC dipole) and of the South Vietnamese upwelling, hereafter referred
to as the SVU, that develops over four main areas. First, the convergence of two gyres creates an eastward jet
departing from the southern part of the central coast of Vietnam between 11°N and 12°N. This convergence gives
rise to an Ekman current-induced coastal upwelling (Dippner et al., 2007; Chen et al., 2012), hereafter referred
as SCU. Second, Ekman pumping-induced upwelling (OFU hereafter) develops offshore in the area of strong
positive wind and surface current vorticity (Liu et al., 2012; Da et al., 2019; Ngo and Hsin, 2021). Third, recent
studies have revealed that coastal upwelling (NCU hereafter) can develop along the northern part of the central
Vietnamese coast (Da et al., 2019; Ngo and Hsin, 2021; To Duy et al., 2022). Last, To Duy et al. (2022) showed
for the first time that upwelling develops off the Mekong Delta in the wake of Con Dao islands (see Fig. 1b, MKU
hereafter). SVU participates to the nutrient enrichment of the surface layer, hence plays an important role in the
biological productivity and in the halieutic resources of the region (Bombar et al., 2010; Liu et al., 2012; Loick-
Wilde et al., 2017; Lu et al., 2018; Loisel et al., 2017). Some authors also showed that the SVU may influence the
functioning of local and regional climate (Xie et al., 2003; Zheng et al., 2016). Understanding precisely the
functioning and variability of the SVU and its response to long-term changes is therefore an important issue.



The interannual variability of the SVU has been investigated in numerous previous studies. In the SCU, OFU and
MKU regions, the interannual variability of summer wind intensity is related to and in phase with the intensity of
the summer monsoon, and is the main driver of the interannual variability of upwelling intensity (Wang et al.,
2006; Chen et al., 2014; Li et al., 2014; Da et al., 2019; Ngo and Hsin, 2021; To Duy et al., 2022). ENSO (El Niño
Southern Oscillation) also impacts the upwelling in those regions, due to its influence on summer monsoon wind
(Wang et al., 2006; Kuo et al., 2004; Loick-Wilde et al., 2017; Da et al., 2019). Some studies (Li et al., 2014; Da
et al., 2019) then revealed that ocean intrinsic variability (OIV) influences the interannual variability of the
eastward jet and of the OFU. This influence of OIV is related to the spatial distribution of summer averaged
surface current vorticity associated with eddies: cyclonic (anticyclonic) eddies located in the area of positive wind
stress curl enhance (weaken) the Ekman pumping-induced OFU. The interannual variability of NCU shows a
completely different behavior. Ngo and Hsin (2021) and To Duy et al. (2022) concluded that wind conditions
favorable to SCU, MKU and OFU were unfavorable to NCU, and vice versa. To Duy et al. (2022) moreover
showed that the influence of wind is weaker for NCU than for the other areas, and that the influence of circulation,
in particular of the spatial organization of the strongly chaotic submesoscale to mesoscale circulation that prevails
over the area, is stronger: on a seasonal average, NCU is inhibited when alongshore currents prevail, and enhanced
when offshore circulation prevails.
The daily to intraseasonal variability of the SVU was much less studied. Available studies, all based on satellite
data, focused on the SCU and OFU. Xie et al. (2007) showed that the upwelling in those areas does not develop
smoothly during the summer, but shows a strong intraseasonal variability related to the wind variability and
Madden Julian Oscillation (MJO). They suggested that SVU experiences two to four events of development and
decay during the summer, in response to the wind fluctuations. Isoguchi and Kawamura (2006) and Liu et al.
(2012) confirmed respectively for the period 2000-2002 and for summer 2007 that the MJO is a strong driver of
the events of southwesterly wind intensification within the season and of the resulting upwelling. They moreover
revealed the effect of tropical storms that can reinforce the southwesterly wind hence the SVU.
To better understand the functioning and variability of the SVU, it is therefore necessary to investigate in detail
the functioning of its daily to intraseasonal variability over its four areas of development (SCU, NCU, OFU and
MKU) and to identify the driving factors. Previous studies mentioned above revealed the role of wind for SCU
and OFU, which should be examined for the other areas. The role of OIV in the interannual variability of the
upwelling, related to mesoscale circulation and eddies in the coastal and offshore area, was highlighted for OFU
and suggested for NCU. It should be examined at the intraseasonal scale, and requires an ensemblist approach as
used by Waldman et al. (2017a, 2018). Chen et al. (2012) also showed from idealized simulations that tides and
river plumes could be also involved in SVU variability, however, to our knowledge, only very few models used
for the SVU study included the effect of tides, and none of them investigated their impact.
The present paper focuses on the daily to intraseasonal variability of the SVU over its four areas of development,
examining in particular the role of atmospheric forcing (wind) and of ocean dynamics and its intrinsic variability.
The effect of tides and rivers will be examined in a future study. A fine-resolution realistic model including tides
and already presented and evaluated in To Duy et al. (2022) for the period 2009-2018 is used. Ensemble
simulations with perturbed initial conditions are performed to study the case of summer 2018, which was an
exceptionally strong summer of upwelling for SCU, OFU and MKU (Ngo and Hsin, 2021; To Duy et al., 2022).



The fine-resolution model and ensemble simulations and the definition of study areas, upwelling indicators and
OIV indicators are presented in Part 2. The intraseasonal variability of the oceanic circulation and of the SVU,
including the role of OIV, are examined respectively in Part 3 and Part 4. Results are summarized and future work
is discussed in Part 5.

**2.  Methodology**
**a)  The 3D hydrodynamical ocean model SYMPHONIE**
To Duy et al. (2022) built a fine-resolution configuration of the 3-D ocean circulation model SYMPHONIE
(Marsaleix et al., 2008, 2019) over the Vietnam coastal region (VNC hereafter), based on a horizontal polar grid
with a resolution decreasing linearly seaward, from 1 km at the Vietnamese coast to 4.5 km offshore, and with 50
vertical levels. We use exactly the same configuration here and show VNC domain in Figure 1a,b. The
atmospheric forcing is computed from the 3-hourly output of the European Center for Medium-Range Weather
Forecasts (ECMWF) 1/8° atmospheric analysis, distributed on http://www.ecmwf.int. Initial and lateral ocean
boundary conditions are prescribed from the daily outputs of the global ocean 1/12° analysis PSY4QV3R1
distributed by the Copernicus Marine and Environment Monitoring Service (CMEMS) on
http://marine.copernicus.eu. The implementation of tides follows Pairaud et al. (2008, 2010) and considers the 9
main tidal harmonics, provided by the 2014 release of the FES global tidal model (Lyard et al., 2006). Freshwater
discharge is provided for 36 river mouths. More details about the model, its configuration and the forcings are
provided in To Duy et al. (2022). They performed and evaluated a 10-year simulation over the period 2009-2018,
hereafter called LONG, showing that it reproduces realistically the temporal (seasonal to interannual) and spatial
variability of the SCS ocean dynamics and water masses. In the LONG simulation, a very strong SVU developed
during summer 2018, due in particular to strong July-August wind (To Duy et al., 2022).
**b)  The ensemble**
We performed an ensemble of ten simulations with perturbed initial conditions between January 1st, 2017 and
December 31st, 2018. For that we used ten different initial conditions for temperature, salinity, sea surface
elevation and currents fields. Most of the OIV develops at mesoscale (Serazin et al., 2015, Waldman et al., 2018),
we therefore only perturbed the mesoscale field, following the same methodology as Waldman et al. (2017a,
2017b, 2018): for the ten simulations of the ensemble, the large-scale state of the initial field is identical, and the
small-scale of the initial field state differs. The common large-scale state is equal to the large-scale state of January
1st, 2017 of the LONG simulation, computed using a 100 km low-pass filter. For XX going from 09 to 18, the
small-scale state of January 1$^{st}$, 20XX of the LONG simulation is computed using a 100 km high-pass filter. The
initial state of member XX of the simulation ensemble is then computed by adding this small-scale state to the
common large-scale state.
**c)  Definition of upwelling areas**
Figure 1c,d shows the SST averaged over June-September (JJAS) 2018 for the ensemble average and for OSTIA
(Operational Sea Surface Temperature and Sea Ice Analysis) reanalysis outputs, available at
ftp://data.nodc.noaa.gov/pub/data.nodc/ghrsst/L4/GLOB/UKMO/OSTIA/. Simulated SST is in good agreement



with observations, showing a large area of colder surface water corresponding to the strong SVU that developed
during summer 2018. We show in Figure 1b,c,d the four boxes used by To Duy et al. (2022), that correspond to
the four main areas of SVU development: boxSC and boxNC for respectively the southern (SCU) and northern
(NCU) coastal upwelling, boxOF for the offshore upwelling (OFU), and boxMK for the upwelling offshore the
Mekong delta (MKU).
**d) Indicators of upwelling intensity**
We compute a SST-based upwelling index following exactly the same methodology as Da et al. (2019) and To
Duy et al. (2022). The daily upwelling index $UI_d$ is computed at each point of the study area that verifies
$SST(x,y,t) < T_o$ as:
$$UI_d(x, y, t) = T_{ref} - SST(x, y, t) \text{ for (x,y,t) where SST(x,y,t)<T}_o \qquad (1)$$
The reference temperature $T_{ref} = 29.2°C$ is computed as the SST averaged over JJAS and over boxT$_{Ref}$, the area
east of the upwelling region that is the least impacted by upwelling (see Figure 1c,d), in the LONG simulation.
The threshold temperature under which upwelling occurs, $T_0 = 27.6°C$, is defined as the optimal upwelling
threshold that covers the largest number of upwelling occurrences but avoids to include cold water horizontally
advected between upwelling areas.
For each box $boxN$, the daily upwelling index $UI_{d,boxN}$ is computed as:
$$UI_{d,boxN}(t) = \frac{\iint_{(x,y) in boxN so that SST(x,y,t)<T_0} (T_{ref} - SST(x,y,t)).dx.dy}{A_{boxN}} \qquad (2)$$
where $A_{boxN}$ is the size of $boxN$. The yearly upwelling index $UI_{y,boxN}$ is computed over the JJAS as:
$$UI_{y,boxN} = \frac{\int_{JJAS} UI_{d,boxN}(t)dt}{ND_{JJAS}} \qquad (3)$$
where $ND_{JJAS} = 122$ days is the JJAS duration.
**e) Indicators of OIV impact**
Following Waldman et al. (2018), we introduce two indicators to quantify the impact of OIV on a given variable
$X$, respectively on the mean state and at the daily scale.
The relative intrinsic variability of the temporal mean state of $X$ over a given period is computed as the ratio
between the ensemble standard deviation $\sigma_i$ and ensemble average $m_i$ of the temporal mean $m_t$ of $X$ over this
period:
$$MI(X) = \frac{\sigma_i\left(m_t\left(X(t,i)\right)\right)}{m_i\left(m_t\left(X(t,i)\right)\right)} \qquad (5)$$
It quantifies the impact of OIV on the summer mean of $X$.



For each day of JJAS 2018, the intrinsic contribution to the total temporal variability of *X* is computed as the ratio
between the time-dependent intrinsic variability and the total temporal variability over JJAS 2018:
$$VI\big(X(t)\big) = \frac{\sigma_i\big(X(t,i)\big)}{\sqrt{m_i\big(\sigma_t(X(t,i))^2\big)}}$$    (6)
It quantifies the impact of OIV on the daily variability of *X*.

### 3.    Intraseasonal variability of wind and ocean circulation

The southwest summer monsoon wind blows from June to September over the SCS with three main peaks of
strong northeastward wind (see the daily time series of wind stress averaged over boxOF, boxSC and boxMK,
Figure 2a,b,c): June (9th-18th June, 10 days), July (28th June-18th July, 21 days) and August (1st-13th August, 13
days). Figure 3a,b,c,d shows for each peak the maps of ensemble average of wind stress and wind stress curl, of
surface current speed and of surface current vorticity averaged over the peak period, and the maps of relative
intrinsic variability (MI) of surface current vorticity over the period. A high (low) value of MI of current vorticity
indicates a strong (weak) OIV and a chaotic (stable) circulation. To quantify the strength of the eastward jet, we
calculate the ensemble mean of the average surface current speed through the meridional transect at 109.9°E,
between 9.5 and 12.2°N (see red line on Figure 3a) during the three peaks.
During the June peak, the area of strong positive wind stress curl extends from the coast to ~113°E, with a narrow
meridional coverage (Figure 3a). The ACC dipole is not clearly formed (Figure 3b,c). The weak eastward jet is
located in the south with a maximum speed of 0.5-0.7 m.s$^{-1}$ around 10-11°N, and a mean strength of 0.51 m.s$^{-1}$.
The circulation is stable in the coastal jet area (MI of current vorticity <50%, Figure 3d), but much more chaotic
over most of the offshore area (MI >200%). During the July period, the area of strong positive wind stress curl is
larger than in June (from 10.5°N to 13°N, extending to 112°E). The eastward jet strengthens, with a mean strength
of 0.78 m.s$^{-1}$, and is more in the north, with a speed of about 0.8-1.1m.s$^{-1}$ near 11-12°N. The ACC dipole, with an
anticyclonic (cyclonic) circulation in the south (north), is more pronounced than in June. The circulation is more
stable than in June in the coastal zone and in the cyclonic and anticyclonic areas (MI of current vorticity ~ 100%).
It is less stable in the northeastern region of boxOF, where MI exceeds 200%. During the August period, the area
of strong positive wind stress curl has the largest meridional and zonal extent, to 114°E. The eastward jet is still
stronger, reaching 1.2-1.5 m.s$^{-1}$ around 10.5-11.5°N and a mean strength of 0.88 m.s$^{-1}$. The ACC dipole is also
stronger, with a well-established and large cyclonic gyre. The surface circulation is more stable compared to June
and July, with a larger surface of low MI of current vorticity (<100%) covering boxOF. In September, the summer
monsoon and the large-scale jet/ACC circulation begin to weaken (not shown).

### 4.    Intraseasonal variability of upwelling

Here we examine the upwelling intraseasonal variability and its intrinsic variability for each upwelling area. Table
1 shows for the four areas the value of yearly upwelling index $UI_{y,boxN}$ for each member and for the ensemble
average, and its relative intrinsic variability $MI(UI_{y,boxN})$. It also shows the values of the correlation coefficients



193 between the daily time series of the ensemble mean of $UI_{d,boxN}$ and of the wind stress components and intensity.

194 Figure 2 shows for each upwelling area the daily time series of wind stress, of $UI_{d,boxN}$ for each simulation and for

195 the ensemble average, and of $VI(UI_{d,boxN})$. Figure 3e,f shows for each summer monsoon peak identified in Section

196 2 the maps of $UI_d$ on the day of maximum $UI_{d,boxN}$ over each peak period and the maps of its relative intrinsic

197 variability $MI(UI_d)$.

### 1. The southern coastal upwelling (SCU)

199 For SCU, $UI_{d,boxSC}$ time series show the same daily chronology for each member and for the ensemble mean

200 (Figure 2b). SCU begins to develop during the first half of June, lasts during the whole summer with a strong

201 intraseasonal variability, and disappears during the first half of September. We obtain three peaks of similar

202 intensity, near June 19th, July 15th and August 16th, in phase with the wind forcing over the area: the correlation

203 between the time series of $UI_{d,boxSC}$ and of the daily averaged wind stress intensity over boxSC is equal to 0.64

204 ($p<0.01$, Table 1). The correlation with the wind stress eastward component, i.e. the component nearly parallel to

205 the south coast, that favors the SCU, reaches 0.71 ($p<0.01$).

206 Over the summer, $VI(UI_{d,boxSC})$ varies between 10% when SCU is weak and 40% during periods of strong SCU,

207 showing similar values for the three upwelling peaks (Figure 2e). The yearly upwelling index $UI_{y,boxSC}$ shows a

208 weak ensemble standard deviation (7% relative to the mean, Table 1). This intrinsic variability of the SCU summer

209 strength is much weaker than its interannual variability: in the 2009-2018 LONG simulation analyzed by To Duy

210 et al. (2022), $UI_{y,boxSC}$ shows a 53% interannual standard deviation relative to its interannual mean. SCU develops

211 in the same area for the ten members, in the coastal zone of the ACC dipole convergence, as shown by the very

212 low values of $MI(UI_d)$ (<50%) over this area (Figure 3e,f). Higher $MI(UI_d)$ values are obtained at the periphery of

213 this area, along the northern and southern flanks of the eastward jet. They are related to the variability of the

214 meridional position of the jet: a jet located further north (south) induces a SCU further north (south).

215 SCU daily to intraseasonal variability is therefore mostly driven by the wind. The OIV mainly results from the

216 meridional position of the jet (that does not vary much), thus affects the SCU at a second order at the intraseasonal

217 scale.

### 2. The Mekong delta shelf upwelling (MKU)

219 For MKU, time series of $UI_{d,boxMK}$ are almost identical for each member and for the ensemble mean (Figure 2c).

220 They show a strong intraseasonal variability, with three peaks of varying intensity following the three wind peaks.

221 The July peak is the strongest, followed by the August peak, then the June peak. The daily chronology of MKU

222 also strongly follows the wind chronology, with a correlation of 0.65 ($p<0.01$) with the wind stress intensity

223 averaged over MKU, and of 0.59 (0.61) with the wind stress eastward (northward) component (Table 1).

224 MKU is very weakly influenced by the OIV: $VI(UI_{y,boxMK})$ never exceeds 30% (Figure 2e) and $MI(UI_{y,boxMK})$ is

225 equal to 6% (Table 1). Again, this intrinsic variability of MKU summer strength is negligible compared to its

226 interannual variability: the interannual standard deviation of $UI_{y,boxMK}$ is equal to 85% in the LONG simulation

227 (To Duy et al., 2022). Spatially, MKU is also very stable. As shown by To Duy et al. (2022), it develops along

228 the northeastward current, in the wake of Con Dao islands (Figures 1, 3e). For the 3 periods of MKU development,



Figure 3d,f shows extremely very weak values of MI both for the surface current vorticity and for the spatial
upwelling index. The circulation is therefore extremely stable in this area, explaining the spatial stability of MKU.
The daily chronology and intensity of MKU are thus mainly driven by the wind, and its position is determined by
non-varying factors, presumably bathymetry, that still need to be investigated, and not by chaotic factors like
(sub)mesoscale circulation. As a result, MKU is hardly affected by OIV.

### 3. The offshore upwelling (OFU)

Again, the daily chronology of OFU is very similar for the ten members and the ensemble mean (Figure 2a), and
in phase with the wind chronology (correlation of 0.65, p<0.01 with the daily wind stress intensity over boxOF,
Table 1). However, contrary to SCU, the intensity of OFU peaks varies throughout the season, though wind stress
intensity is similar during those peaks. We obtain two strong peaks (~1.0°C) in the heart of summer on July 19[th]
and August 13[th], a moderate peak (~0.6°C) at the end of August, and two small peaks (~0.2°C) at the beginning
and end of summer, on June 18[th] and September 16[th]. $VI(UI_{d,boxOF})$ also varies a lot seasonally, and is maximum
and much stronger than for SCU and MKU during OFU peaks (Figure 2e): it reaches 90% for the July peak, 70%
during the August peaks, and respectively 30% and 50% during the small June and September peaks. On the
summer average, $MI(UI_{y,boxOF})$ is equal to 18% (Table 1), again stronger than for SCU and MKU, but still much
lower than the interannual variability (126%, To Duy et al., 2022). The regional daily wind stress therefore drives
the daily to intraseasonal variability of OFU at the first order. However, OIV also significantly influences this
daily variability, and this influence varies seasonally.
To understand the mechanisms that explain the intraseasonal variability of OFU intensity, we examine its
functioning during its three main peaks: June, July and August. Da et al. (2019) and To Duy et al. (2022) showed
that OFU is mainly induced by Ekman pumping and develops in the area of strong positive wind stress curl and
current vorticity. The eastward jet and ACC dipole that favor the development of OFU are much stronger and well
established in the heart of summer than at the beginning and end of the summer monsoon (Figure 3bc and Part 3):
in June and September, contrary to July and August, the area of positive current vorticity over boxOF is indeed
very small and not located over an area of strong positive wind curl, not favoring Ekman pumping (Figure 3e).
The intraseasonal variability of OFU peaks is thus explained by the intraseasonal variability of large-scale
circulation.
We then examine the mechanisms that explain why the July and August peaks show different intrinsic variability
but similar ensemble mean of OFU intensity (Figure 2a). Figure 4a shows the maps of $UI_d$ on the day of maximum
$UI_{d,boxOF}$ over the July OFU peak period and the maps of average surface current vorticity during the this period
for 2 members of strong OFU (13, maximum $UI_{d,boxOF}$ = 1.53 °C; 17, maximum $UI_{d,boxOF}$ = 1.42 °C, Figure 2a)
and 2 members of weak OFU (15, maximum $UI_{d,boxOF}$ = 0.77 °C; 18, maximum $UI_{d,boxOF}$ = 0.65 °C). In July, the
eastward jet is much stronger than in June (Figure 3b,c and Part3). OFU develops mainly in the area of positive
wind stress curl and current vorticity along the northern flank of the jet (Figures 3c,e and 4). When the position
of the eastward jet is south, as for members 13 and 17, the area where positive current vorticity and positive wind
curl coincide is larger than average, favoring Ekman pumping. This results in a stronger OFU, with a larger
extension to the northeast. This is the opposite when the position of the eastward jet is north (members 15 and
18). Figure 3d,f shows strong values of MI of current vorticity and upwelling intensity (> 100%) along the



eastward jet and in the northeast area of boxOF. This confirms that OFU intrinsic variability in July is related to
the effect of eastward jet meridional position variability on the circulation and on the upwelling that develops
along the northern flank of the jet. Figure 4b shows the maps of $UI_d$ on the day of maximum $UI_{d,boxOF}$ over the
August OFU peak period and the maps of average surface current vorticity and average wind stress curl during
this period for 2 members of strong OFU (14, maximum $UI_{d,boxOF}$ = 1.65 °C; 13, maximum $UI_{d,boxOF}$ = 1.20 °C,
Figure 2a) and 2 members of weak OFU (10, maximum $UI_{d,boxOF}$ = 0.82 °C; 16, maximum $UI_{d,boxOF}$ = 0.89 °C).
In August, part of OFU still develops in the area of positive surface current vorticity along the northern flank of
the eastward jet, but to a lesser extent than in July (Figures 3e, 4). The meridional position of the jet does not vary
a lot from one member to another (Figure 4b), as confirmed by the lower values of MI of current vorticity in the
jet area (Figure 3d). The eastward jet is thus stronger and more stable than in July (Figure 3b,c and Part 3), and
does not induce a strong intrinsic variability of OFU. Instead, August OFU mainly develops in the area of positive
vorticity north of the jet associated with the cyclonic eddy of the ACC dipole (Figures 4b,3e). The variations of
zonal position of this eddy explain the variability of OFU intensity. From members 14, to 13, 10 and 16, this eddy
is located more and more to the east, i.e. further and further away from the area of strong positive wind stress curl,
resulting in a weaker and weaker OFU (Figures 4b). The variability of circulation in the northern part of boxOF
therefore explains OFU intrinsic variability in August. This variability is moreover lower than in July: MI of
current vorticity and of $UI_d$ in this northern part (highlighted by the black triangle in Figure 3d,e,f) is lower in
August than in July. The more stable jet in August, that results in a smaller intrinsic variability of OFU along the
jet, and the smaller intrinsic variability of current vorticity in the northern cyclonic part, where OFU mostly
develops, therefore explain the intrinsic variability of OFU in August and the fact that it is smaller than in July.
**4.  The northern coastal upwelling (NCU)**
The ten members and the ensemble mean simulate NCU with a strong intraseasonal variability and a similar
chronology (Figure 2d), completely different from the chronology obtained for the three other areas. A strong
NCU develops at the beginning of the summer (from June 10th to July 4th, reaching ~1.2°C for the ensemble
average), and a weak NCU develops at the end of August (August 26th to 31st, reaching ~0.2°C). During those
periods, NCU chronology follows the wind chronology for the ten members: $UI_{d,boxNC}$ peaks correspond to peaks
of northward (i.e. alongshore) wind favorable to NCU, around June 18th and 25th, July 2nd, and August 28th and
29th. There still a significant correlation between the time series of $UI_{d,boxNC}$ and the time series of wind stress
northward component over boxNC, that favors the Ekman upwelling in this area (0.37, p<0.01, Table 1). It is
however much weaker than correlations obtained for the other areas (at least 0.64). Moreover, although northward
wind peaks occur during the whole summer, NCU does not develop from mid-July to mid-August.
NCU shows the strongest OIV of the four areas. $VI(UI_{d,boxNC})$ reaches 170% during the June-July peak, and is
much smaller during the rest of the summer, reaching at most ~80% during the short late August peak (Figure 2e).
The strong OIV in June explains the strong OIV at the summer scale (Table 1): $MI(UI_{s,boxNC})$ is equal to 37%,
twice larger than for OFU and ~6 times larger than for SCU and MKU. It is of the same order of magnitude as the
interannual variability of MKU summer strength, even if twice smaller (72% in the LONG simulation, To Duy et
al., 2022).



These results suggest that the daily to intraseasonal chronology of upwelling in boxNC is partly driven by wind,
but to a lesser extent than in the other areas, and that other factors are involved that induce a strong intrinsic
variability of NCU both at the daily and summer scales. To identify those factors, we examine three periods: the
period of strong wind over boxNC and strong NCU in June (June 10th – July 4th), the period of strong wind but no
NCU in July (July 17th - 22nd), and the period of strong wind and weak NCU at the end of August (August 26th -
31st). We show in Figure 5 the maps of ensemble average of average surface current, current vorticity and MI of
current vorticity over each period, and the maps of $UI_d$ and $MI(UI_d)$ on the day of maximum $UI_{d,boxNC}$ over each
period.
During the June event, ensemble average circulation in and around boxNC is globally offshore oriented. This
favors the Ekman transport, hence the development of NCU. The ensemble spreading of NCU strength is very
strong: $MI(UI_d)$ spatially reaches 500% in the eastern part of boxNC (Figure 5e), and $VI(UI_{d,boxNC})$ reaches 170%
(Figure 2e). Figure 6 shows the maps of wind speed and curl, current speed and vorticity and $UI_d$ on the day of
maximum $UI_{d,boxNC}$ over the June wind peak period for two members of strong NCU (10 and 16) and two members
of weak NCU (09 and 11). A cyclonic gyre in the north and anticyclonic gyre in the south meet in boxNC for
members 10 (between 13°N and 14°N) and 16 (between 14°N and 15°N). This induces a convergence and an
offshore current resulting in a strong upwelling, following the same mechanism as for SCU. For members 09 and
11, cyclonic and anticyclonic gyres do not meet in boxNC, but either north or south of boxNC, not inducing
offshore oriented current over boxNC. Instead a weak NCU is induced by a favorable northward alongshore
current in the northern part of boxNC (Figure 6). As already observed at the interannual scale by To Duy et al.
(2022) for the interannual variability, mesoscale circulation of strongly chaotic nature in and around boxNC
therefore drives the NCU development and explains its high intrinsic variability during the June period.
Between mid-July and mid-August, the large-scale circulation (ACC dipole and eastward jet) is strongly
established (see Parts 3 and 4c). The western part of the cyclonic eddy covers boxNC and induces a strong
southward alongshore current over this region. Close to the coast, this alongshore current is associated with a
divergent circulation, hence with a coastward component and a coastal downwelling which inhibits the NCU (see
the high negative vorticity in this area in Figure 5b). This large-scale ocean circulation is common for the ten
members and systematically prevents the NCU to develop (see the weak MI of current vorticity and the weak
upwelling in Figure 5). This explains the very weak $UI_{d,boxNC}$ and $VI(UI_{d,boxNC})$ during this period (Figure 2e).
During the August event, the average circulation is similar to the mid-July and mid-August circulation described
above (Figure 5a,b). However, with the weakening of the summer monsoon, the ACC dipole structure
progressively weakens, the negative vorticity is less strong and the current is a bit more offshore oriented: NCU
is not as strong as in June, but it can develop easier than in the middle of the summer. $VI(UI_{d,boxNC})$ consequently
increases, but stays smaller than in June.
Our results therefore show that the development of NCU at the intraseasonal scale is first related to the large-scale
circulation (eastward jet + ACC dipole). When this circulation is strongly established during the heart of summer,
it prevents the NCU from developing. NCU is only allowed to develop at the beginning and end of summer, when
this circulation is weaker. Then, inside those periods of "allowed NCU development", the daily chronology and





the intensity of NCU are first driven by the alongshore component of wind, but also by the organization of
mesoscale circulation in and around boxNC that explains its very strong intrinsic variability.

### 5. Conclusion and future work

An ensemble of ten fine-resolution simulations with perturbed initial conditions was performed and analyzed in
this paper to represent and investigate the daily to intraseasonal variability of ocean circulation and of SVU over
its different areas of development and the influence of OIV.
The ensemble was used to examine the seasonal variability and intrinsic variability of the circulation in the SVU
region during summer 2018. In June, the eastward jet is weak and mainly located in the south, the ACC dipole is
not formed and the circulation is only stable in the coastal area. In July, the jet is stronger and the ACC dipole is
clearly formed. The circulation in the area of positive vorticity north of the jet is more stable. In August southwest
monsoon wind is the strongest and has the largest area of influence, inducing an even stronger eastward jet and a
pronounced ACC dipole. The circulation is stable over a larger area than in July.
We then examined the seasonal variability and intrinsic variability of the upwelling.
For SCU, MKU and OFU, the daily chronology and intraseasonal variability of the upwelling are quite similar
and mainly driven by the summer monsoon wind, with upwelling maxima in phase with strong southwest wind
periods. Their intrinsic variability is much weaker than their interannual variability.
The development of MKU and SCU is related to ocean circulation processes, respectively the northeastward
current and eastward jet, that are spatially quite stable. As a result, MKU and SCU show a very weak intrinsic
variability, both at the daily scale and on average over the summer (lower than 10%), both spatially and on average
over the area. Peaks of OIV are related to peaks of upwelling intensity. SCU develops as long as wind conditions
are favorable over the area of the ACC dipole convergence, that does not vary much spatially. MKU develops
along the northeastward current that flows offshore the Mekong delta and is spatially very stable.
OFU shows a stronger intrinsic variability, both at the daily scale and for the summer average (18%), both spatially
and on average over boxOF. The seasonal variability of OFU intensity and intrinsic variability are not only driven
by the wind, but also related to the period of the season. The large-scale circulation (ACC dipole and eastward
jet) that enhances the Ekman pumping – induced OFU is weak in June and September, whereas it is strongly
established in the heart of summer in July and August. This explains the stronger OFU intensity and OIV during
the July-August period. Moreover, for similar OFU intensity, the impact of OIV is weaker in August than in July.
In July, OFU mainly develops along the northern flank of the eastward jet. The meridional position of this jet is
quite variable, explaining the strong intrinsic variability of the July OFU: a southern (northern) position of the jet
induces a larger (smaller) common area between positive curl of wind stress and current, hence induces a stronger
(weaker) upwelling. In August, this position is much more stable. Moreover, OFU mainly develops in the area of
cyclonic activity north of the jet, related to mesoscale eddies of strong chaotic nature. OFU intensity depends on
the zonal position of this cyclonic activity with respect to the wind curl: a cyclonic eddy (not) located in the area



of strong positive wind stress curl results in stronger (weaker) OFU. The circulation in this area is moreover more
stable than in July, explaining the weaker intrinsic variability in August.
NCU daily chronology and the intraseasonal variability is completely different from the other areas, and its
intrinsic variability is much higher. At the intraseasonal scale, the development of NCU is driven by the large-
scale circulation. During the heart of summer, from mid-July to end of August, the eastward jet and ACC dipole
are strongly established, inducing a strong southward alongshore current over boxNC which annihilates NCU,
explaining its very low intensity and intrinsic variability. At the beginning and, to a smaller extent, end of summer,
the large-scale ACC dipole structure and associated eastward current are weaker, allowing offshore current to
develop. Inside these periods, NCU chronology is driven by the wind but also by the development (or not) of
offshore oriented currents related to the spatial organization of coastal eddies. NCU shows a strong intrinsic
variability related to the strong chaotic variability of those small mesoscale structures, of the same order as its
interannual variability (37%).
We investigated here the role of OIV on the circulation and upwelling in the SVU region. Further studies are now
required to investigate the influence of other factors, including tides and rivers. We moreover developed an ocean-
atmosphere regional coupled model that will allow to study the impact of upwelling on atmosphere and climate,
at local and regional scales, that was studied until now using satellite data and an atmosphere-only model (Xie et
al. 2003; Zheng et al., 2016; Yu et al., 2020). The impact of upwelling on the marine ecosystem should also be
studied, using for example the dynamical-biogeochemical coupled model developed by Ulses et al. (2016) and
Herrmann et al. (2017). Finally, the long-term evolution of the upwelling should be studied, in particular since
Herrmann et al. (2020,2022) showed that summer monsoon winds may weaken in response to climate change.
Last, the upwelling that develops offshore the Mekong Delta, MKU, was revealed by To Duy et al. (2022) and
confirmed by the present study, and the northern coastal upwelling, NCU, was revealed by Da et al. (2019) and
confirmed by Ngo and Hsin (2021). Extremely few in-situ observations are available over their areas and periods
of development, and field campaigns are therefore necessary to better understand their functioning. This study
explored the impact of OIV on the South Vietnam Upwelling, but is more generally of interest for the scientific
community working on the functioning and variability of upwellings and on the effect and modeling of OIV.

**Code and data availability**

The SYMPHONIE model is available on the webpage of the SIROCCO group, https://sirocco.obs-mip.fr/. Sea
surface temperature, currents and windstress simulated by the ensemble over summer 2018 are freely available
on https://doi.org/10.5281/zenodo.7504087.

**Authors Contribution**

Marine Herrmann, To Duy Thai and Claude Estournel designed the experiments and To Duy Thai carried them
out. Marine Herrmann prepared the manuscript with contributions from all co-authors.



**Competing interests**
The authors declare that they have no conflict of interest.

**Acknowledgements**
This work is a part of LOTUS international joint laboratory (lotus.usth.edu.vn). PhD studies of To Duy Thai were
funded through an IRD ARTS grant and a "Bourse d'Excellence" from the French Embassy in Vietnam.
Numerical simulations were performed using CALMIP HPC facilities (project P13120) and the cluster OCCIGEN
from the CINES group (project DARI A0080110098). This paper is a contribution to celebrate the 100 years
Anniversary of the Institute of Oceanography, Vietnam Academy of Science and Technology.

**Tables**
Table 1 : For each upwelling area: value of the yearly upwelling index $UI_{y,boxN}$ for each member of the ensemble,
of the ensemble mean $mi(UI_{y,boxN})$ and of $MI(UI_{y,boxN})$, and correlation coefficients between the daily times
series of the ensemble mean of the daily upwelling index $UI_{d,boxN}$ and of the wind stress eastward and northward
components and intensity. Only correlations associated with p-values <0.01 are shown.

| Members | 09 | 10 | 11 | 12 | 13 | 14 | 15 | 16 | 17 | 18 | $mi(UI_{y,boxN})$ (°C) | $MI(UI_{y,boxN})$ (%) | Correlation between $m_i(UI_{d,boxN})$ and wind stress eastward component | Correlation between $m_i(UI_{d,boxN})$ and wind stress northward component | Correlation between $m_i(UI_{d,boxN})$ and wind stress intensity |
|---|---|---|---|---|---|---|---|---|---|---|---|---|---|---|---|
| BoxOF | 0.32 | 0.35 | 0.26 | 0.39 | 0.37 | 0.39 | 0.26 | 0.26 | 0.42 | 0.31 | 0.33 | 18 % | 0.62 | 0.62 | 0.65 |
| BoxSC | 1.25 | 1.35 | 1.25 | 1.49 | 1.39 | 1.42 | 1.27 | 1.38 | 1.48 | 1.29 | 1.36 | 7 % | 0.60 | 0.71 | 0.64 |
| BoxNC | 0.11 | 0.24 | 0.08 | 0.15 | 0.17 | 0.14 | 0.25 | 0.25 | 0.26 | 0.13 | 0.18 | 37 % | -- | 0.37 | -- |
| BoxMK | 0.18 | 0.18 | 0.19 | 0.19 | 0.17 | 0.18 | 0.17 | 0.18 | 0.2 | 0.17 | 0.18 | 6 % | 0.59 | 0.61 | 0.65 |




**Figures**

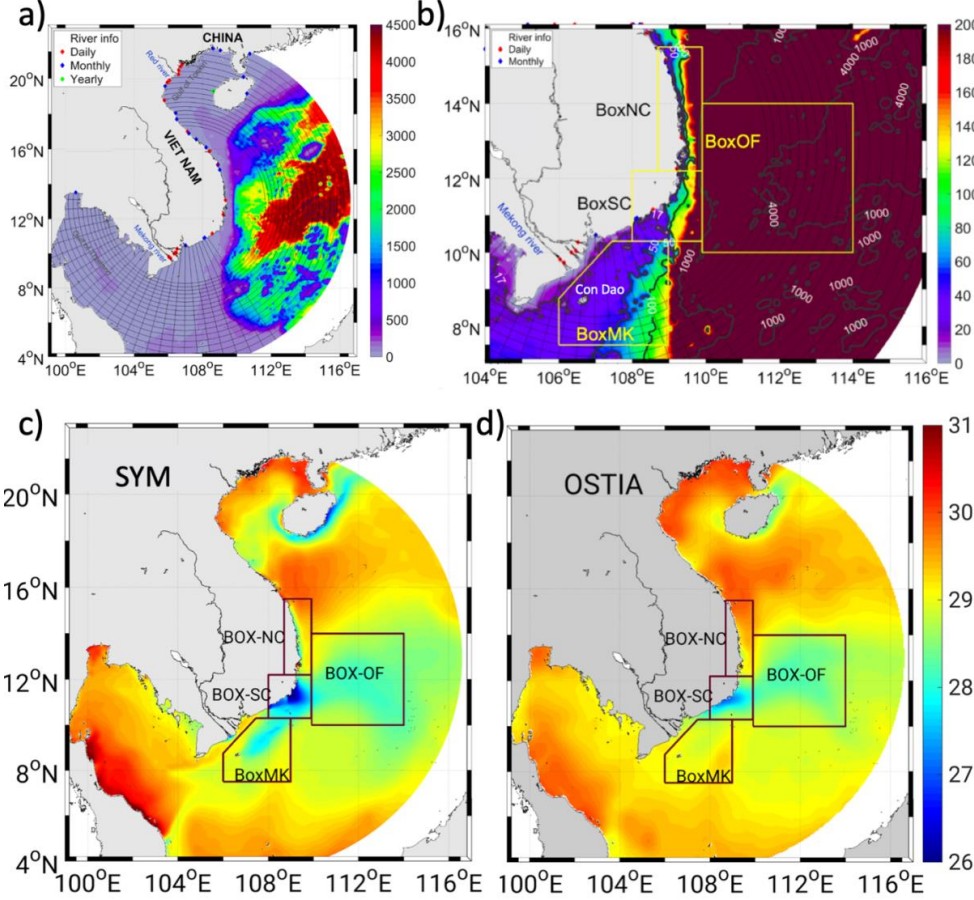

Figure 1: (a) Characteristics of the orthogonal curvilinear computational grid (black lines, not all the mesh points are shown for visibility purposes) and bathymetry (colors, meter, *GEBCO_2014*) used for the VNC configuration of the SYMPHONIE model. Dots show the location of rivers for which we used daily (red), monthly (blue) and yearly climatology (green) discharge values (see To Duy et al. 2022 for more details). (b) Bathymetry (meter) over the SVU region. The 4 boxes used for the study of SVU are displayed in yellow. (c,d) SST (°C) averaged over JJAS 2018 computed from the SYMPHONIE ensemble average (c) and from OSTIA reanalysis (d).



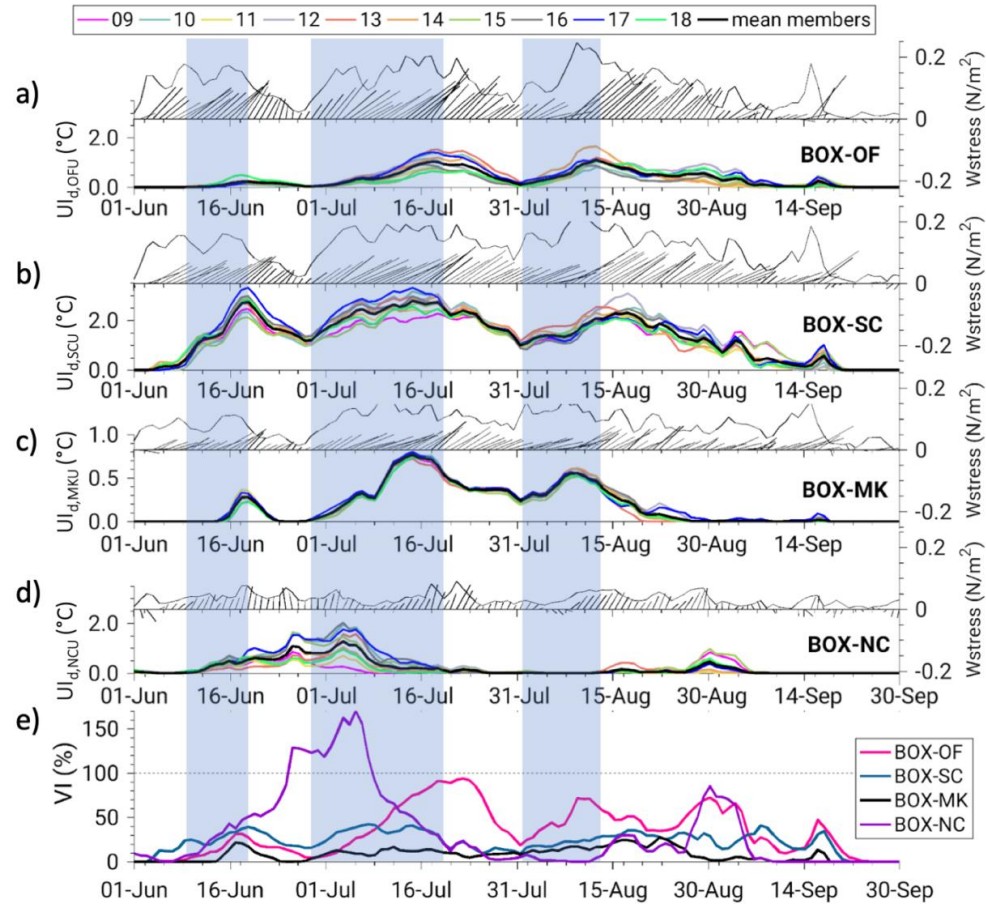

440

Figure 2: (a,b,c,d) Daily time series between June 1$^{st}$ and September 30$^{th}$ 2018 of direction (arrows) and intensity

(black line) of spatially averaged wind stress (N.m$^{-2}$) over each upwelling area, and time series of $UI_{d,boxN}$ for each

simulation (colored lines) and for the ensemble average (black thick line) for each upwelling area (a, BoxSC; b,

BoxOF; c, BoxMK; d, BoxNC). (e) Daily time series of $VI(UI_{d,boxN})$ for each upwelling area. Periods of southwest

monsoon wind peaks are highlighted in blue.






Figure 3: Maps of ensemble average of average wind stress (a, arrows, N.m$^{-2}$) and wind stress curl (a, colors, N.m$^{-}$
$^{3}$), of average surface current speed (b, m.s$^{-1}$) and vorticity (c, s$^{-1}$), of $UI_d$ (e, °C) on the day of maximum $UI_{d,boxOF}$
over each wind peak period (left June, middle July and right August), and maps of relative intrinsic variability
(MI) of average surface current vorticity (d, %) and of $UI_d$ (f, %) over each period. Black lines: 3.10$^{-7}$ N.m$^{-3}$ iso-
contours of average wind stress curl. Red segment (a): meridional transect at 109.9°E, 9.5 - 12.2°N used to
compute the eastward jet strength. Arrows (b-f): average surface current during each period. Black triangles (c-f):
area of high current vorticity northern of the eastward jet during the July and August periods. Pink lines (e, f):
isobaths (meters).

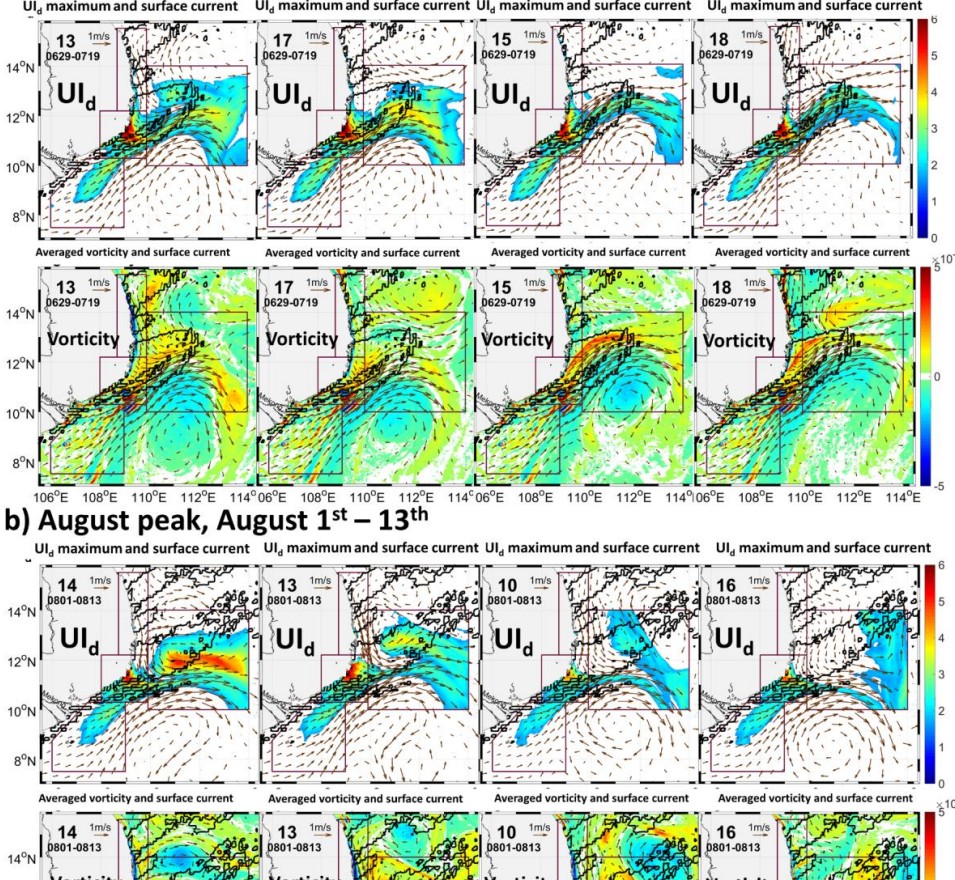

Figure 4: Maps of $UI_d$ (top, °C) on the day of maximum $UI_{d,boxOF}$ over the (a) July and (b) August peaks of OFU, and of average surface current vorticity (bottom, s$^{-1}$) during each peak for 2 members of strong OFU (members 13 and 17 for July, members 14 and 13 for August, Figure 2a) and 2 members of weak OFU (members 15 and 18 for July, members 10 and 16 for August). Black contours: 3.10$^{-7}$ N.m$^{-3}$ iso-contours of average wind stress curl during each period. Arrows: average surface current (m.s$^{-1}$).



466

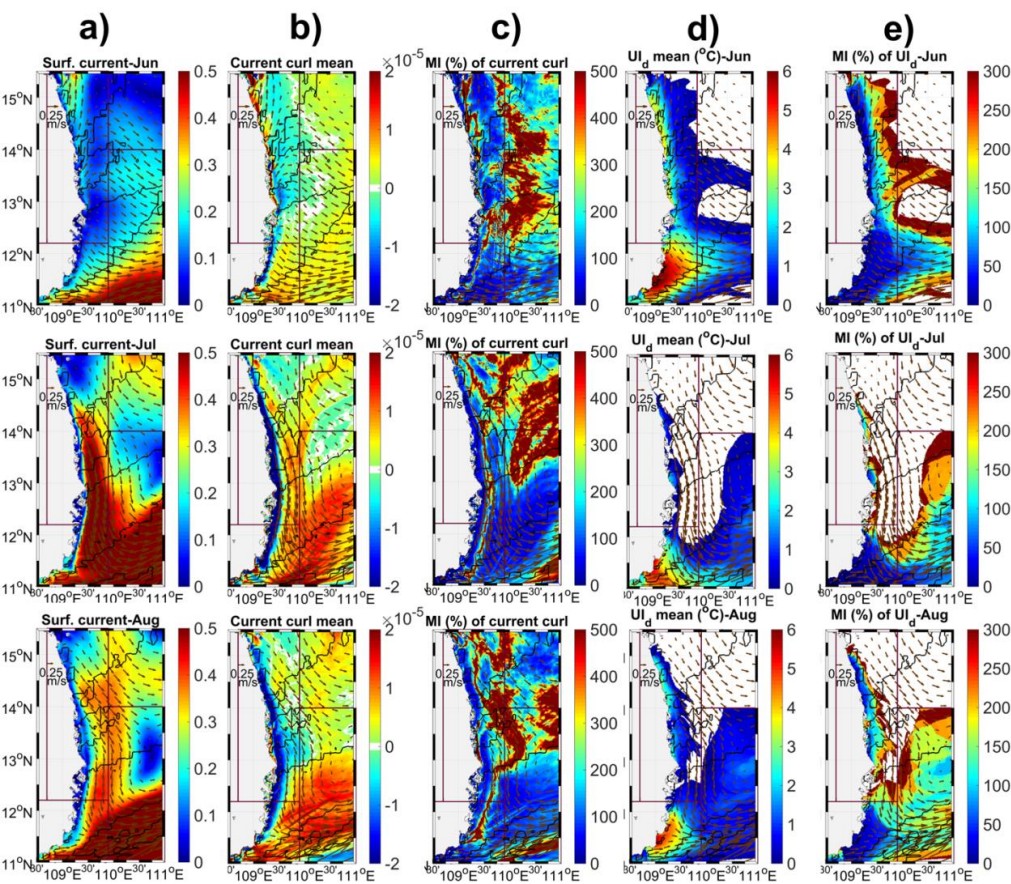

467

Figure 5 : Maps of ensemble average of average surface current speed (a, m.s$^{-1}$) and vorticity (b, s$^{-1}$) over each
wind peak period over BoxNC (1st row June 10th – July 4th, 2nd row July 17th - 22nd and 3rd row August 26th - 31st),
of $UI_d$ (d, °C) on the day of maximum $UI_{d,boxNC}$ over each period, and of relative intrinsic variability MI of average
surface current vorticity (c, %) and $UI_d$ (e, %) over each period. Black contours: 3.10$^{-7}$ N.m$^{-3}$ iso-contours of
average wind stress curl during each period. Arrows: average surface current (m.s$^{-1}$).



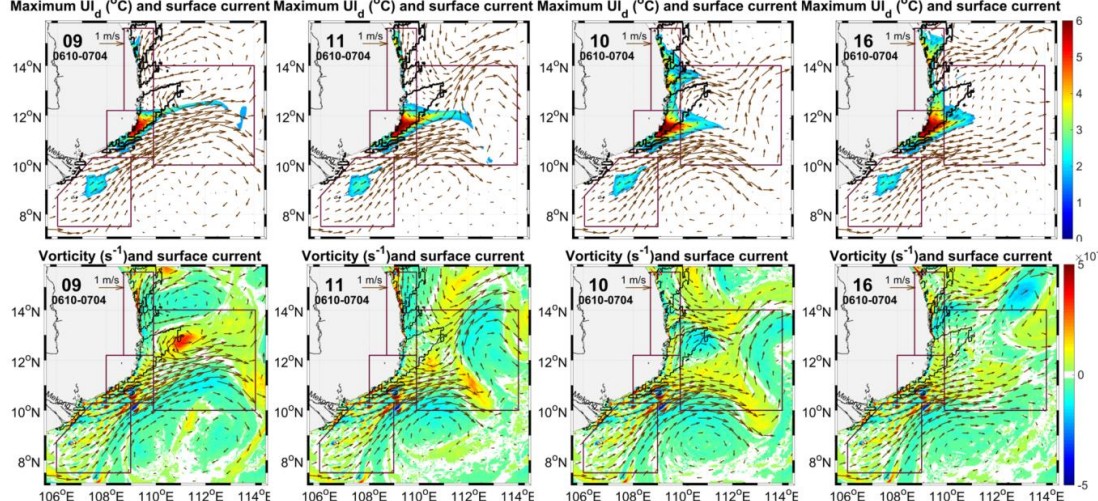

Figure 6 : Maps of $UI_d$ (1st row, °C) on the day of maximum $UI_{d,boxNC}$ over the July wind peak period over BoxNC and maps of average surface current vorticity (2nd row, s⁻¹) for 2 members of weak NCU (9, 11) and 2 members of strong NCU (10, 16) during the June-July period. Black contours: $3.10^{-7}$ N.m⁻³ iso-contours of average wind stress curl during each period. Arrows: average surface current (m.s⁻¹).



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
