# Peer review of "Intraseasonal variability of the South Vietnam Upwelling, South"

_EGUsphere, 2022_

## Referee Comment (RC1)

Comments on the paper OS-2022-1443 "Intraseasonal variability of the South Vietnam Upwelling, South China Sea: influence of atmospheric forcing and ocean intrinsic variability" by Marine Herrmann, Thai To Duy, Claude Estournel

In this paper, the authors investigate the variability of the South Vietnam Upwelling (SVU) at time scales from day to season and propose a method for estimating the contribution of the ocean intrinsic variability (OIV) to the total variability of SUV. This work is an extension of the previous work, by the same authors, that has been focused on larger time scales of SUV variability. I reviewed the previous paper submitted to Ocean Science Journal in 2022, and recently published, and I appreciated the results on the whole.

The method of investigation of the SVU variability in both studies is based on numerical simulations using SYMPHONIE model which was extensively validated in the previous study. In the present study, the authors employ the approach of ensemble simulations with perturbations of initial conditions. Small scale (spatial scale) perturbations of initial conditions were generated for initializing the model fields. These perturbations are assumed to represent a spectrum of ocean circulation variability. A total of ten simulations, one year long each, were performed to obtain a range of variability which is supposed to represent the real OIV occurring under atmospheric conditions of one particular year chosen for analysis. By applying the proposed method, the authors demonstrate that the contribution of OIV varies significantly in time, during summer months, and in space, in four sub-domains of the SVU.

I find that the method used in this study is original. It provides valuable results which extend the model-based characterization of SVU variability that the authors got started in their previous study. I think that the findings presented in this paper are of considerable interest and worth publishing in Ocean Science Journal.

However, there are numerous minor grammar mistakes that did not disrupt the flow of the article but should be corrected. I provide below a list of recommendations that I hope will be of some help.

Additionally, I would like the following points to be addressed.

- How the tree periods of strong wind forcing (L166) were identified? Two of them start at low wind and end at high wind period of time. In contrast, the first and shortest period of time comprises only dates with high wind speed. How the presented results can be sensitive to the length of time intervals?
- I suggest to provide a better explanation of indicators defined in section 2d,e. For example, what does the "yearly upwelling index UI" mean? What the authors want to show using this indicator? In addition, the word "yearly" is misleading because the time average is calculated over four months of the year.
- What reason did the authors follow to choose the acronyms MI and VI? What does "M", "V", and "I" means? The acronyms do not match the quantities defined in section 2, L155 and L160.
- In L152, the authors introduce the indicator VI for a variable X. They should precise what variables are targeted: temperature is evident, but what else? How MIs for other variables covary? Do they follow a trend similar to UI?

Minor comments and suggestions

- Numbering of sections should be corrected. For example, section 1 (L198) appears after section 4 (L189). Subsections a,b,c,d of section 2 should be changed to 2a, 2b, … .

L28: NCU shows a behavior different from that revealed in three other upwelling areas.

L29: a large-scale

L30: preventing NCU development.

L37: remove "over the region"

L38: please put (AC) after anticyclonic , and (C) after cyclonic.

L39: South Vietnam (please use this name throughout the whole text).

L64-68: Cut this sentence in two. … for NCU than for three other upwelling regions. New sentence: In contrast, the influence of circulation, in particular, the spatial organization of (remove "the strongly") chaotic … was found to be stronger: ….

L76: a comma is missed before hence.

L87: Assuming "… (wind) and intrinsic variability of ocean dynamics".

Ln88,100: 3D or 3-D ? please harmonize.

LN100: VNC does not match "the Vietnam coastal region". Please correct.

L113: July- August wind : please precise the wind direction. Put 'due' after 'in particular'.

L114: I would suggest "Ensemble simulations".

L139: box T_ref is not shown in Fig. 1.

L161: "contribution" is better than "impact"

L172: remove "see", correct "in Figure 3a.

L175: "speed" instead of "strength"

L178: with a mean "speed"

L183: zonal "extension"

L184: Assuming: jet is stronger (or strongest?) with velocity reaching … and the mean current speed up to 0.9 m/s.

186: "area" instead of "surface"

L199: Assuming "similar" , not "the same". Some difference can exist.

L215: OIV "is related to"

216: I would remove parentheses: position of the jet which does not vary much, thus affecting …

L220: They "also" show

L228: Why "in the wake" and not "behind"? Usually the wake has the maximum extension of 8-10 times the size of the object that generated the wake. Is this the case? Please precise.

L233: I find the last statement "hardly affected by OIV" not coherent with considerations given in L231-232. Please verify and correct.

L246: the word "seasonally" means from one season to another. It is not that the authors mean. Please correct.

L252-255: The text needs clarification. The Ekman pumping velocity depends on the curl of the wind stress. If the curl is large enough the Ekman pumping velocity is large. If not, some other factors come into play. A better (more clear and physically consistent) explanation of weak upwelling is required.

L261: what is part 3?

L263: "south" of what?  And also put "the" before "positive".

L265: extension "toward" the northeast. The text in L262-266 should be reworded for clarity.

L274: here and in other places: "to less extent".

L278: Remove "The", start with "Variations in zonal …"

L285: remove "the" before "smaller"

L290: remove "the" before "summer"

Ln300: … to 37% that is twice larger than …

L301-302: Should chose between: " same order" and "two times smaller" which are not equal.

L305: I would suggest to put a dot after "areas" and start a new sentence with : Therefore other factors induce …

L314: It is not possible to see 500% level in Fig. 5. The scale is limited by 300%.

L326: what is part 3 and 4c? Fig 4c?

L337-340: this paragraph contains generalities without providing mechanisms of NCU development. It should be revised.

L341: Assuming " by the alongshore wind component which is northward from 10 of June until the end of August.

L353: "In August" the circulation is stable over larger area than in July.

L350-360: show a very weak (less than 10%) intrinsic variability in space and time, both at daily scale and on average over summer.

L364: OFU shows stronger intrinsic variability (18%), both at daily scale and for the whole summer period, and also in space.

L378: … different from "that found in three" other areas …

L386: … related to strong chaotic variability of mesoscale structures with the order of magnitude similar to interannual variability (37%). If "small" is used, the scale should be clearly defined.

L388: the "effect" of OIV …

L398: over these two areas.

---

## Referee Comment (RC2)

Comments and suggestions on the research article OS-2022-1443 *"Intraseasonal variability of the South Vietnam Upwelling, South China Sea: influence of atmospheric forcing and ocean intrinsic variability"* - authors *Marine Herrmann, To Duy Thai and Claude Estournel.*

The main objective of this article is to understand the mechanism of a well-known phenomenon in the Southeast Asian waters during summer: the upwelling at the Southern Vietnam coast (SVU) in the South China Sea (SCS). This manuscript is the continuation of recently published work in Ocean Science :To Duy et al., 2022 that studied the interannual variability of the SVU. In this article, authors - using the same numerical configuration as the previous paper - studied the influence of summer monsoon wind and ocean intrinsic variability on the four main areas of the SVU: the northern coastal upwelling (NCU), the southern coastal upwelling (SCU), the offshore upwelling (OFU) and the shelf off the Mekong River mouth (MKU), in daily and intraseasonal time scale. Results are extracted from ten simulations on the same period (2017 - 2018) with different initial conditions in temperature, salinity, currents and sea surface height (perturbations only made on mesoscale fields). Summer 2018 (June, July, August) is chosen for the case study analyses.

I find this article very well structured and written. The scientific objectives are presented concisely followed by a clear explanation of applied methods and rigorous analyses of the result. The scientific question and computational method are original. However, I have a few questions and suggestions for the authors in order to better understand their work.

In the methodology part (part 2), the bathymetry database used for the model's grid is not defined. Since the resolution of bathymetry is important in the study of current fields, it should be mentioned in the methodology part. Did the authors use a fine resolution bathymetry for the coastal zone?

The second question concerns the ocean intrinsic variability (OIV). If I understand correctly, this OIV represents an ensemble of ocean intrinsic properties such as temperature, salinity, currents, etc. In the authors' opinion, which parameter(s) are the main factor(s) controlling this OIV?

The third question is in regard to the application of the research findings on predicting the upwelling development. Could it be possible to predict the intensity of the upwelling zone using wind forecast?

The final question relates to the role of tide and river discharge on the variability of the upwelling intensity. In the authors' opinion, how important are tidal currents at coastal upwelling zones like NCU and SCU? How much influence do the enormous discharges from the Mekong river have on the properties of MKU?

Technical corrections: some parts need to be re-numbered
Line 198: it should be 4.1
Line 218: It should be 4.2
Line 234: It should be 4.3
Line 287: It should be 4.4

To conclude, the work provided in this paper meets the quality requirements of Ocean Science and is worth publishing. I greatly appreciate the effort of the authors helping us better understand the mechanism of the SVU.

---

## Author Comment (AC1)

Comments on the paper OS-2022-1443 "Intraseasonal variability of the South Vietnam Upwelling, South China Sea: influence of atmospheric forcing and ocean intrinsic variability" by Marine Herrmann, Thai To Duy, Claude Estournel, by Alexei Sentchev.

In this paper, the authors investigate the variability of the South Vietnam Upwelling (SVU) at time scales from day to season and propose a method for estimating the contribution of the ocean intrinsic variability (OIV) to the total variability of SUV. This work is an extension of the previous work, by the same authors, that has been focused on larger time scales of SUV variability. I reviewed the previous paper submitted to Ocean Science Journal in 2022, and recently published, and I appreciated the results on the whole.

The method of investigation of the SVU variability in both studies is based on numerical simulations using SYMPHONIE model which was extensively validated in the previous study. In the present study, the authors employ the approach of ensemble simulations with perturbations of initial conditions. Small scale (spatial scale) perturbations of initial conditions were generated for initializing the model fields. These perturbations are assumed to represent a spectrum of ocean circulation variability. A total of ten simulations, one year long each, were performed to obtain a range of variability which is supposed to represent the real OIV occurring under atmospheric conditions of one particular year chosen for analysis. By applying the proposed method, the authors demonstrate that the contribution of OIV varies significantly in time, during summer months, and in space, in four sub-domains of the SVU.

I find that the method used in this study is original. It provides valuable results which extend the model-based characterization of SVU variability that the authors got started in their previous study.

I think that the findings presented in this paper are of considerable interest and worth publishing in Ocean Science Journal.

However, there are numerous minor grammar mistakes that did not disrupt the flow of the article but should be corrected. I provide below a list of recommendations that I hope will be of some help.

Additionally, I would like the following points to be addressed.

We warmly thank Alexei Sentchev for the time and attention devoted to our paper, and for those positive and constructive comments. We have carefully considered all his comments and suggestions in the revised version of our manuscript. In what follows, and in the highlighted version of the manuscript, our answers and modifications are highlighted in blue. Line numbers refer to the highlighted version of the revised manuscript.

1) How the tree periods of strong wind forcing (L166) were identified? Two of them start at low wind and end at high wind period of time. In contrast, the first and shortest period of time comprises only dates with high wind speed. How the presented results can be sensitive to the length of time intervals?

This comment highlighted the need to better explain how those periods were chosen. The purpose of those periods is to understand, for a given period and area of upwelling peak, which factors govern the development of upwelling. Those periods hence need to cover the period over which the upwelling develops, i.e. during which $UI_{d,boxN}$ goes from a value 0 to a maximum value. Using periods shorter than the upwelling development period, for example only the day of maximum, or a few days around the maximum, produced less meaningful results, it is therefore important to consider the full period of upwelling development.

The periods defined at the beginning of section 3 and highlighted in blue in Figure A and Figure 2 of the revised paper cover the 3 peaks of OFU development : June (9th-18th June, 10 days), July (28th June-18th July, 21 days) and August (1st-13th August, 13 days). They also roughly cover the SCU and MKU development period. For OFU, MKU and SCU, the chronology is indeed quite similar and driven by wind, and our results were quite robust to the periods of investigation, as long as they roughly

covered the upwelling development period. For OFU and SCU, there is a range of ~3 days for the day of maximum depending on the member of the ensemble (see Figure A below), but our tests showed that using specific maximum days for each member or one common maximum day did not significantly change our results. The multiplicity of periods, for each area and each member, would have made the explanations complicated to follow. For the sake of readability of the manuscript, we therefore used the same periods defined above and highlighted in blue in Figure A below for all members and for the 3 areas.

As shown in Section 4.4 of the manuscript, NCU follows a completely different chronology than the three other areas. We therefore used in Section 4.4 and Figures 5 and 6 of the manuscript different periods than for OFU, SCU and MKU to study NCU development (and annihilation) and intraseasonal variability. Those periods, now highlighted in green in Figure A below and in the revised Figure 2, are all periods of northeastward wind over BoxNC, but with different NCU answers : NCU development in June (June 10$^{th}$ – July 4$^{th}$) and to a less extent in August (row August 26$^{th}$ - 31$^{st}$), and no NCU development in July (July 17$^{th}$ - 22$^{nd}$).

[Figure]

*Figure A: (a,b,c,d) Daily time series between June 1$^{st}$ and September 30$^{th}$ 2018 of direction (arrows) and intensity (black line) of spatially averaged wind stress (N.m$^{-2}$) over each upwelling area, and time series of $UI_{d,boxN}$ for each simulation (colored lines) and for the ensemble average (black thick line) for each upwelling area (a, BoxSC; b, BoxOF; c, BoxMK; d, BoxNC). (e) Daily time series of $IV_d(UI_{d,boxN})$ for each upwelling area. Periods to study OFU, SCU and MKU development are highlighted in blue, and periods used to study MKU development in green.*

→ Following this comment, we explained into more details in the revised manuscript how the periods were chosen (Section 3 lines 176-188 for OFU, SCU and MKU and section 4.4 lines 322-337 for NCU) and highlighted them in green and blue (as explained above) in Figure 2 of the revised manuscript. We also corrected captions of Figures 3 to 6.

2) I suggest to provide a better explanation of indicators defined in section 2d,e. For example, what does the "yearly upwelling index UI" mean? What the authors want to show using this indicator? In addition, the word "yearly" is misleading because the time average is calculated over four months of the year.

Upwelling index indicators are used to assess and quantify the intensity of upwelling (and its variability) at the daily scale, and integrated over the summer:

- $UI_d(x,y,d)$ quantifies the intensity of upwelling at day $d$ over a given point $(x,y)$.

- $UI_{d,boxN}(d)$ quantifies the daily intensity of upwelling at day $d$, integrated over the region *boxN*.

- $UI_{JJAS,boxN}$ quantifies the daily intensity of upwelling at day d, integrated over the region boxN. The subscript $y$ was indeed not very meaning full and we replaced $UI_y$ by $UI_{JJAS}$

→ We modified section 2.3 to explain that more clearly in the revised version of the paper (lines 141-157).

3) What reason did the authors follow to choose the acronyms MI and VI? What does "M", "V", and "I" means? The acronyms do not match the quantities defined in section 2, L155 and L160.

We initially chose the acronyms MI and VI following Waldman et al. (2018) but we agree that they are not really meaningfull. We therefore changed them, using the acronym $IV_d$ for the indicator of effect of **I**ntrinsic **V**ariability at **D**aily scale, and the acronym $IV_{tm}$ for the indicator of effect of **I**ntrinsic **V**ariability at **T**emporal **M**ean scale.

→ We modified this everywhere in the text, in Table 1 and in Figures 2, 3 and 5.

Moreover, our explanation of the meaning of those indicators was not completely clear, we therefore completely modified the text of section 2.5 (lines 159-174). We hope that it is now clearer.

4) In L152, the authors introduce the indicator VI for a variable X. They should precise what variables are targeted: temperature is evident, but what else? How MIs for other variables covary? Do they follow a trend similar to UI?

$X(t,i)$ at time $t$ and for ensemble member $i$ can be any variable characterizing the ocean circulation and upwelling intensity in the simulation member $i$: temperature, salinity, sea surface elevation, current speed or vorticity, upwelling index. It can depend on space (e.g. SST, current vorticity) or not (e.g. $UI_{d,boxN}$ and $UI_{JJAS,boxN}$ ). Here we investigate the effect of OIV on surface circulation and on upwelling, we therefore apply those indicators on the upwelling indices ($UI_d$, $UI_{d,boxN}$ and $UI_{JJAS,boxN}$) and on the surface current vorticity and examine their relationship.
→ This was better explained in the revised version of the manuscript (lines 160-162 and 173-174).

$UI_d$ is directly related to the SST (equation 1, $UId(x,y,t) = T_{Ref} – SST(x,y,t)$). In the area of OFU, MKU and SCU, $IV_{tm}$ of UId and SST shows similar spatial trend as $IV_{tm}$ of the surface current vorticity, as shown by Fig. 3d,f. This actually allowed us to explain throughout the text why some areas were strongly (OFU), or almost not (SCU and MKU), impacted by OIV : areas of strong current vorticity chaotic variability are the areas were the upwelling is strongly influenced by OIV.
→ We acknowledged this in the revised version of the manuscript, lines 314-316.

For NCU the situation is a bit different : in June, during the period of NCU development, the impact of OIV on UId is smaller at the coast and larger when going further offshore, but it is not the case of the vorticity (Figure 5). Upwelling develops at the coast for most members of the ensemble, but is offshore extension varies among members and explains NCU intrinsic variability, as shown in Figure 6. This offshore extension is actually related to the direction of the currents : alongshore (northward or southward) currents prevent NCU from extending offshores, whereas offshore oriented currents favor it. Over BoxNC, the upwelling intrinsic variability is therefore not really related to the intrinsic variability of current vorticity, but to the intrinsic variability of current direction relative to the coast.

→ We acknowledged this in the revised version of the manuscript, lines 354-356

Minor comments and suggestions

- Numbering of sections should be corrected. For example, section 1 (L198) appears after section 4 (L189). Subsections a,b,c,d of section 2 should be changed to 2a, 2b, … .

→ There was indeed a problem with the numbering, it was fixed in the revised version of the manuscript.

L28: NCU shows a behavior different from that revealed in three other upwelling areas.

→ corrected (taking into account the 300 words limit for the abstract)

L29: a large-scale

→ corrected

L30: preventing NCU development.

→ corrected

L37: remove "over the region"

→ corrected

L38: please put (AC) after anticyclonic , and (C) after cyclonic.

→ corrected

L39: South Vietnam (please use this name throughout the whole text).

→ corrected

L64-68: Cut this sentence in two. … for NCU than for three other upwelling regions. New sentence: In contrast, the influence of circulation, in particular, the spatial organization of (remove "the strongly") chaotic … was found to be stronger: ….

→ corrected

L76: a comma is missed before hence.

→ corrected

L87: Assuming "… (wind) and intrinsic variability of ocean dynamics".

→ This sentence was not clear and was corrected (lines 90-91). Here we examine the role of wind, and the role of ocean dynamics, including their chaotic part (using perturbed initial conditions ensembles).

Ln88,100: 3D or 3-D ? please harmonize.

→ corrected using 3-D

LN100: VNC does not match "the Vietnam coastal region". Please correct.

→ We explained in the manuscript that VNC stands for VietNam Coast (line 104)

L113: July- August wind : please precise the wind direction. Put 'due' after 'in particular'.

→ corrected (line 118)

L114: I would suggest "Ensemble simulations".

→ corrected

L139: box T_ref is not shown in Fig. 1.

→ We corrected this and Figure 1 now shows boxT$_{Ref}$.

L161: "contribution" is better than "impact"

→ corrected (lines 159 and 169)

L172: remove "see", correct "in Figure 3a.

→ corrected

L175: "speed" instead of "strength"

→ corrected

L178: with a mean "speed"

→ corrected

L183: zonal "extension"

→ corrected

L184: Assuming: jet is stronger (or strongest?) with velocity reaching … and the mean current speed up to 0.9 m/s.

→ corrected , lines 200 and 205-206

186: "area" instead of "surface"

→ corrected

L199: Assuming "similar" , not "the same". Some difference can exist.

→ corrected

L215: OIV "is related to"

→ corrected

216: I would remove parentheses: position of the jet which does not vary much, thus affecting …

→ corrected

L220: They "also" show

→ corrected

L228: Why "in the wake" and not "behind"? Usually the wake has the maximum extension of 8-10 times the size of the object that generated the wake. Is this the case? Please precise.

Indeed strictly speaking we did not want to refer to the wake, but to the area of Con Day island located downstream of the northeastward current. So "behind" is more correct.

→ corrected

L233: I find the last statement "hardly affected by OIV" not coherent with considerations given in L231-232. Please verify and correct.

→ Here we used "hardly" as a synonym of "almost not", but since it was not clear, we corrected that.

L246: the word "seasonally" means from one season to another. It is not that the authors mean. Please correct.

→ Indeed, it was corrected to intraseasonally

L252-255: The text needs clarification. The Ekman pumping velocity depends on the curl of the wind stress. If the curl is large enough the Ekman pumping velocity is large. If not, some other factors come into play. A better (more clear and physically consistent) explanation of weak upwelling is required.

Our results show that the intraseasonality of OFU is driven by the wind, and further enhanced by the large scale circulation (eastward jet and AC/C dipole). Northeastward wind intensity is slightly stronger in July and August than in June and September (Figure 2) with a larger area of positive wind curl (Figure 3a). Moreover, positive current vorticity developing in BoxOF in the area of positive wind curl is much higher in July and August, which further enhances Ekman pumping (Figure 3e). The intraseasonal variability of OFU peaks is thus explained by the intraseasonal variability of wind and large-scale circulation

→ Following this comment, we reworded those lines (274-278.), we hope it is clearer now

L261: what is part 3?

We meant Section 3 of the paper → corrected to "Section 3 above"

L263: "south" of what? And also put "the" before "positive".

We meant more to the south than the average position → corrected

L265: extension "toward" the northeast. The text in L262-266 should be reworded for clarity.

When the position of the eastward jet is more to the south than average, as for members 13 and 17 (see Figure 4 of the paper), the area of positive current vorticity in BoxOF, northern of the jet, is well located in the area of positive wind curl. This combination of positive current vorticity and positive wind curl enhances the Ekman pumping induced upwelling (Figure 4). This results in a strong OFU covering a large area. This is the opposite when the position of the eastward jet is located more to the north than average (members 15 and 18, Figure 4) : the area of positive current vorticity located in the positive wind curl region is smaller, not enhancing the Ekman pumping induced upwelling (Figure 4).

→ Following this comment, we reworded those lines (286 to 292), we hope it is clearer now

L274: here and in other places: "to less extent".

→ corrected

L278: Remove "The", start with "Variations in zonal …"

→ corrected

L285: remove "the" before "smaller"

→ corrected

L290: remove "the" before "summer"

→ corrected

Ln300: … to 37% that is twice larger than …

→ corrected

L301-302: Should chose between: " same order" and "two times smaller" which are not equal.

→ We used only "twice smaller" (lines 332-333).

L305: I would suggest to put a dot after "areas" and start a new sentence with : Therefore other factors induce …

→ corrected

L314: It is not possible to see 500% level in Fig. 5. The scale is limited by 300%.

→ The sentence was replaced by "exceeds 300%", line 344

L326: what is part 3 and 4c? Fig 4c?

→ this was replaced by "Section 3", line 356

L337-340: this paragraph contains generalities without providing mechanisms of NCU development. It should be revised.

→ This paragraph was simplified and shortened, lines 370-373

L341: Assuming " by the alongshore wind component which is northward from 10 of June until the end of August.

→ Following the comment above, this paragraph was simplified and shortened, lines 370-373

L353: "In August" the circulation is stable over larger area than in July.

→ corrected

L350-360: show a very weak (less than 10%) intrinsic variability in space and time, both at daily scale and on average over summer.

→ corrected

L364: OFU shows stronger intrinsic variability (18%), both at daily scale and for the whole summer period, and also in space.

→ corrected

L378: … different from "that found in three" other areas …

→ corrected

L386: … related to strong chaotic variability of mesoscale structures with the order of magnitude similar to interannual variability (37%). If "small" is used, the scale should be clearly defined.

→ corrected

L388: the "effect" of OIV …

→ corrected

L398: over these two areas.

→ corrected

---

## Author Comment (AC2)

Comments and suggestions on the research article OS-2022-1443 "Intraseasonal variability of the South Vietnam Upwelling, South China Sea: influence of atmospheric forcing and ocean intrinsic variability" - authors Marine Herrmann, To Duy Thai and Claude Estournel.

The main objective of this article is to understand the mechanism of a well-known phenomenon in the Southeast Asian waters during summer: the upwelling at the Southern Vietnam coast (SVU) in the South China Sea (SCS). This manuscript is the continuation of recently published work in Ocean Science :To Duy et al., 2022 that studied the interannual variability of the SVU. In this article, authors - using the same numerical configuration as the previous paper - studied the influence of summer monsoon wind and ocean intrinsic variability on the four main areas of the SVU: the northern coastal upwelling (NCU), the southern coastal upwelling (SCU), the offshore upwelling (OFU) and the shelf off the Mekong River mouth (MKU), in daily and intraseasonal time scale. Results are extracted from ten simulations on the same period (2017 - 2018) with different initial conditions in temperature, salinity, currents and sea surface height (perturbations only made on mesoscale fields). Summer 2018 (June, July, August) is chosen for the case study analyses.

I find this article very well structured and written. The scientific objectives are presented concisely followed by a clear explanation of applied methods and rigorous analyses of the result. The scientific question and computational method are original. However, I have a few questions and suggestions for the authors in order to better understand their work.

We warmly thank the reviewer for the time and attention devoted to our paper, and for those positive and constructive comments. We have carefully considered all the comments and suggestions in the revised version of our manuscript. In what follows, and in the highlighted version of the manuscript, our answers and modifications are highlighted in green. Line numbers refer to the highlighted version of the revised manuscript.

1) In the methodology part (part 2), the bathymetry database used for the model's grid is not defined. Since the resolution of bathymetry is important in the study of current fields, it should be mentioned in the methodology part. Did the authors use a fine resolution bathymetry for the coastal zone?

The numerical grid was built in autumn 2018 based on the latest release of GEBCO bathymetry dataset, i.e. the GEBCO_2014 dataset released in April 2015 at a 30 seconds interval (~0.9 km) and available from www.gebco.net. The resolution of GEBCO_2014 was suitable for this since our grid resolution varies from a minimum of 1026 m at the coast to 4435 m offshore (see fig. A below).

→ This information was added in the revised version of our paper (lines 106-107 of the highlighted manuscript).

[Figure]

Fig. A : mesh size (meter) of the VNC grid

2) The second question concerns the ocean intrinsic variability (OIV). If I understand correctly, this OIV represents an ensemble of ocean intrinsic properties such as temperature, salinity, currents, etc. In the authors' opinion, which parameter(s) are the main factor(s) controlling this OIV?

Ocean Intrinsic variability (OIV), as opposed to the forced variability, corresponds to the unpredictable part of ocean variability: it is the variability that is not induced by the variability of external forcing factors (atmospheric forcing, lateral boundary conditions, river flow), but by the chaotic behavior of ocean dynamics. It indeed affects all variables that characterize the state of the ocean (temperature, salinity, currents, etc). Most studies have shown that mesoscale to submesoscale structures are the major source of OIV (see Penduff et al., 2011; Serazin et al., 2016; Waldman et al., 2018; Da et al. 2019; as references). The results of our study are in line with these conclusions, as we show that for the two areas most affected by the OIV, i.e., the OFU, and even more so the NCU, the chaotic part of the circulation and upwelling variability is related to the small-scale structures that develop in these regions.

→ We added a few lines in the introduction to better define the OIV at the beginning, and in the conclusion (lines 60-63 and 418-420) to highlight the contribution of our study to the understanding of OIV).

3) The third question is in regard to the application of the research findings on predicting the upwelling development. Could it be possible to predict the intensity of the upwelling zone using wind forecast?

The results of To-Duy et al. 2022 and this paper showed that MKU and SCU have very low intrinsic variability on the intraseasonal scale, and their interannual variability is mainly determined by the summer wind intensity over the SVU region. This suggests that upwelling development in these coastal regions could be largely predicted using wind forecasts, provided of course that these forecasts are reliable.

OFU has stronger intrinsic variability, but its daily to interannual variability is also primarily determined by the intensity of summer wind stress and the curvature of wind stress. Our results therefore suggest that at first order it can be partially predicted from wind forecasts, but with uncertainties in its daily to summer integrated magnitude that can vary by a factor of 1.5 to 2 (see Figure 2 and Table 1 of the paper).

The intrinsic variability of NCU is much higher. On interannual and intraseasonal scales, its development is driven by the large-scale circulation: conditions that favor upwelling in the other three areas prevent NCU. Moreover, during periods potentially favorable for NCU, the role of wind is less important than for the other areas, and the sub-mesoscale to mesoscale chaotic circulation that prevails over the coastal zone has a strong impact on NCU development. For NCU, wind forecasts could therefore be used to identify favorable and unfavorable periods for NCU development. However, favorable periods would be associated with a high uncertainty rate in terms of NCU intensity, given the strong impact of OIV that could result in very strong but also extremely weak NCUs.

→ We added a paragraph in the conclusion (420-428) to highlight this potential application of our findings for the SVU predictability.

4) The final question relates to the role of tide and river discharge on the variability of the upwelling intensity. In the authors' opinion, how important are tidal currents at coastal upwelling zones like NCU and SCU? How much influence do the enormous discharges from the Mekong river have on the properties of MKU?

[Figure]

Fig. B : Co-Tidal charts of M2, K1, S2, O1 (colors show the tidal amplitude in m) from Phan et al. (2019).

Previous modeling studies suggested that tides could influence the SVU development (Chen et al. 2012). Figure B shows the amplitude and phase charts for the four principal tidal components over the South China Sea: M2, K1, S2, and O1 (Phan et al. 2019). Tides are overall very weak in the relatively deep northern coastal and offshore regions. Their amplitude is larger over the shallow Mekong shelf region, with amplitudes of each component reaching ~1 m, suggesting that tidal currents could be strong in this area. Indeed, residual currents over this area can reach 5 cm/s, as shown by the map of tidal residual currents produced by the FES2022 tidal atlas (see Fig. C below).

[Figure]

Fig. C : residual tidal currents simulated by FES_2022 tidal atlas, courtesy of Damien Allain, LEGOS.

The main rivers on the Vietnamese coast are the Mekong and Red Rivers. The Red River is quite far from the SVU region, but the Mekong plume could affect the region through its effect on vertical stratification as well as horizontal density gradient, especially the MKU region that directly receives water from the Mekong. This rapid analysis suggests that tides and rivers should have a minor impact on OFU, but may potentially affect MKU, and possibly SCU and NCU. A detailed analysis of these effects on these four upwelling areas is underway and will be presented in a paper to be submitted soon.

→ Results concerning the effects of tides and rivers on the SVU over its different areas of development will be presented in this coming paper, but we added a sentence in the manuscript regarding this potential effect (lines 429-432).

5) Technical corrections: some parts need to be re-numbered

Line 198: it should be 4.1

Line 218: It should be 4.2

Line 234: It should be 4.3

Line 287: It should be 4.4

→ There was indeed an overall numbering problem in the paper that was fixed in the revised version.

To conclude, the work provided in this paper meets the quality requirements of Ocean Science and is worth publishing. I greatly appreciate the effort of the authors helping us better understand the mechanism of the SVU

Once again warmly thank the reviewer for this very nice comment that is much appreciated.

**References**

Chen, C., Z. Lai, R. C. Beardsley, Q. Xu, H. Lin, and Viet, N. T.: Current separation and upwelling over the southeast shelf of Vietnam in the South China Sea, J. Geophys. Res., 117, C03033, doi:10.1029/2011JC007150, 2012

Da, N. D., Herrmann, M., Morrow, R., Niño, F., Huan, N. M., and Trinh, N. Q.: Contributions of Wind, Ocean Intrinsic Variability, and ENSO to the Interannual Variability of the South Vietnam Upwelling: A Modeling Study, J. Geophys. Res. Ocean., 124, 6545–6574, https://doi.org/10.1029/2018JC014647, 2019.

Penduff, T., Juza, M., Barnier, B., Zika, J., Dewar, W. K., Treguier, A. M., Molines, J. M., and Audiffren, N.: Sea level expression of intrinsic and forced ocean variabilities at interannual time scales, Journal of Climate, 24, 5652–5670. https://doi.org/10.1175/JCLI-D-11-00077.1, 2011

Phan, H. M., Ye, Q., Reniers, A. J.H.M., Stive, M. J. F: Tidal wave propagation along The Mekong deltaic coast, Estuarine, Coastal and Shelf Science, 220, 73-98, https://doi.org/10.1016/j.ecss.2019.01.026; 2019

Sérazin, G., Meyssignac, B., Penduff, T., Terray, L., Barnier, B., and Molines, J. M.: Quantifying uncertainties on regional sea level change induced by multidecadal intrinsic oceanic variability, Geophys. Res. Lett., 43, 8151–8159, https://doi.org/10.1002/2016GL069273, 2016.

To Duy, T., Herrmann, M., Estournel, C., Marsaleix, P., Duhaut, T., Bui Hong, L., and Trinh Bich, N.: Role of wind, mesoscale dynamics and coastal circulation in the interannual variability of South Vietnam Upwelling, South China Sea. Answers from a high resolution ocean model, Ocean Sci., 18, 1131–1161, https://doi.org/10.5194/os-18-1131-2022, 2022

Waldman, R., Somot, S., Herrmann, M., Sevault, F., and Isachsen, P. E.: On the Chaotic Variability of Deep Convection in the Mediterranean Sea, Geophys. Res. Lett., 45, 2433–2443, https://doi.org/10.1002/2017GL076319, 2018.

---

## Author Response (AR2)

To :
John Huthnance, Editor of Ocean Science

From :

Marine Herrmann, Claude Estournel
LEGOS, Toulouse, France

To Duy Thai
Institute of Oceanography, Nha Trang, Vietnam

Toulouse, March 17th, 2023

Dear Editor,

We warmly thank you for your work with our manuscript "Intraseasonal variability of the South Vietnam Upwelling, South China Sea: influence of atmospheric forcing and ocean intrinsic variability".

We have done all the "technical corrections" that you recommended in the revised and final version of our manuscript.

Best regards

Marine Herrmann, coordinating author